# LOGIC-LM: Empowering Large Language Models with Symbolic Solvers for Faithful Logical Reasoning

**Liangming Pan**     **Alon Albalak**     **Xinyi Wang**     **William Yang Wang**

University of California, Santa Barbara

{liangmingpan, alon_albalak, xinyi_wang, wangwilliamyang}@ucsb.edu

## Abstract

Large Language Models (LLMs) have shown human-like reasoning abilities but still struggle with complex logical problems. This paper introduces a novel framework, LOGIC-LM, which integrates LLMs with symbolic solvers to improve logical problem-solving. Our method first utilizes LLMs to translate a natural language problem into a symbolic formulation. Afterward, a deterministic symbolic solver performs inference on the formulated problem. We also introduce a self-refinement module, which utilizes the symbolic solver's error messages to revise symbolic formalizations. We demonstrate LOGIC-LM's effectiveness on five logical reasoning datasets: ProofWriter, PrOntoQA, FOLIO, LogicalDeduction, and AR-LSAT. On average, LOGIC-LM achieves a significant performance boost of 39.2% over using LLM alone with standard prompting and 18.4% over LLM with chain-of-thought prompting. Our findings suggest that LOGIC-LM, by combining LLMs with symbolic logic, offers a promising avenue for faithful logical reasoning. [1]

## 1 Introduction

Logical reasoning is a cognitive process that involves using evidence, arguments, and logic to arrive at conclusions or make judgments (Huang and Chang, 2023). It plays a central role in intelligent systems for problem-solving, decision-making, and critical thinking. Recently, large language models (LLMs) (Brown et al., 2020; Ouyang et al., 2022a; OpenAI, 2023) have exhibited emergent ability to "reason" like human (Wei et al., 2022a). When prompted with step-wise explanations of reasoning ("chain of thoughts"), or a simple prompt "Let's think step by step.", these models are able to answer questions with explicit reasoning steps (Wei et al., 2022b; Kojima et al., 2022).

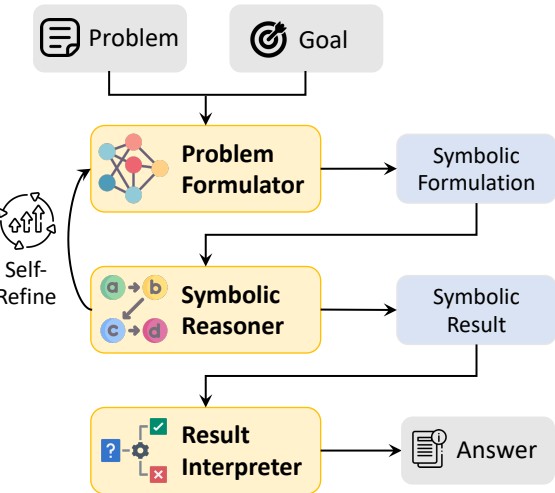

Figure 1: Overview of our LOGIC-LM framework.

Despite the advances of LLMs, they still struggle with complex logical reasoning problems (Liu et al., 2023b). Recent studies (Golovneva et al., 2023; Ribeiro et al., 2023b; Lyu et al., 2023) found that LLMs occasionally make *unfaithful* reasoning, *i.e.*, the derived conclusion does not follow the previously generated reasoning chain. While chain-of-thought may imitate human reasoning processes, the fundamental nature of LLMs remains that of black-box probabilistic models, lacking a mechanism to guarantee the faithfulness of reasoning (Shanahan, 2022). In contrast, *symbolic inference engines*, such as expert systems (Metaxiotis et al., 2002), are faithful and transparent because the reasoning is based on symbolic-represented knowledge and follows well-defined inference rules that adhere to logical principles. The main obstacle is how to accurately translate a problem into symbolic representations, considering the inherent ambiguity and flexibility of natural language. This is precisely where LLMs excel, making LLMs a promising complement to symbolic solvers.

This drives our exploration of neuro-symbolic methods that integrate LLMs with symbolic reasoning. As illustrated in Figure 1, we present LOGIC-

---

[1] Code and data are publicly available at https://github.com/teacherpeterpan/Logic-LLM.

LM, a novel framework that decomposes a logical reasoning problem into three stages: *Problem Formulation*, *Symbolic Reasoning*, and *Result Interpretation*. During problem formulation, an LLM converts the natural language description of the problem into an appropriate symbolic formulation, identifying key entities, facts, and rules present in the problem statement. Subsequently, at the symbolic reasoning stage, a deterministic symbolic solver performs inference on the symbolic formulation. Lastly, a result interpreter explains the output and maps it to the correct answer. By incorporating LLMs with symbolic solvers, we can exploit the robust natural language understanding capabilities of LLMs to precisely represent the problem using symbolic representations, while also taking advantage of the logical faithfulness and transparency offered by symbolic solvers. To improve the accuracy of the symbolic parsing, we also incorporate the idea of self-refinement to iteratively revise the generated logical form using the error messages from the symbolic solver as feedback.

We showcase the adaptability and effectiveness of LOGIC-LM on five logical reasoning datasets: *ProofWriter* (Tafjord et al., 2021), *PrOntoQA* (Saparov and He, 2023), *FOLIO* (Han et al., 2022), *AR-LSAT* (Zhong et al., 2022), and the *LogicalDeduction* dataset from BigBench (Srivastava et al., 2022). These datasets cover a wide range of logical reasoning problems, including:

- Deductive Reasoning problems
- First-Order Logic (FOL) reasoning problems
- Constraint Satisfaction Problems (CSP)
- Analytical Reasoning (AR) problems

We integrate four types of symbolic inference tools tailored to these problems: 1) *logic programming engine* that supports deductive reasoning through forward/backward chaining; 2) *FOL inference engine* that derives new conclusions based on FOL rules and facts, 3) *constraint optimization* engine that provides solvers for CSP over finite domains, and 4) *boolean satisfiability problem (SAT) solver* that solves analytical reasoning problems.

Our evaluations show that the strategy of integrating LLMs with symbolic solvers performs significantly better than purely relying on LLMs for logical reasoning, with an average improvement of 39.2% over the standard prompting and 18.4% over the chain-of-thought prompting (§ 4.1). We also find that LOGIC-LM becomes increasingly effective as the required reasoning depth increases (§ 4.3). Finally, by analyzing the impact of self-refinement, we highlight the effectiveness of incrementally revising symbolic formalizations when interacting with the symbolic solver (§ 4.4).

## 2   Related Work

**Language Models for Logical Reasoning.**   Recent works in adapting LLMs for logical reasoning tasks can be broadly categorized into two groups: 1) *fine-tuning approaches* that optimize LLMs' reasoning ability through fine-tuning or training specialized modules (Clark et al., 2020; Tafjord et al., 2022; Yang et al., 2022), and 2) *in-context learning approaches* that design special prompts to elicit LLMs' step-by-step reasoning capabilities. Typical methods include chain-of-thought prompting (Wei et al., 2022b; Wang et al., 2023) that generates explanations before the final answer and the least-to-most prompting (Zhou et al., 2023) that breaks the problem down into simpler components that can be solved individually. Both the above approaches perform reasoning directly over natural language (NL), providing greater flexibility than symbolic-based reasoning. However, the intrinsic complexity and ambiguity of NL also bring undesired issues such as unfaithful reasoning and hallucinations.

Different from prior works, we use *symbolic language* as the basic unit of reasoning. This effectively transfers the burden of executing complex, precise reasoning from LLMs to more reliable, interpretable external symbolic solvers. Simultaneously, we leverage the strong in-context learning ability of LLMs to formulate the NL-based problem into suitable symbolic representations, thus maintaining the benefit of flexibility.

Although prior works (Mao et al., 2019; Gupta et al., 2020; Manhaeve et al., 2021; Cai et al., 2021; Tian et al., 2022; Pryor et al., 2023) also propose neuro-symbolic methods to combine neural networks with symbolic reasoning, these methods suffer from limitations such as hand-crafted or specialized module designs that are not easily generalizable, or brittleness due to the difficulty of optimization. In contrast, we propose a more generalizable framework that integrates modern LLMs with symbolic logic without the need for training or designing complex problem-specific modules.

**Tool-augmented Language Models.**   Language models have inherent limitations such as the inability to access up-to-date information, take actions, or perform precise mathematical reasoning. To

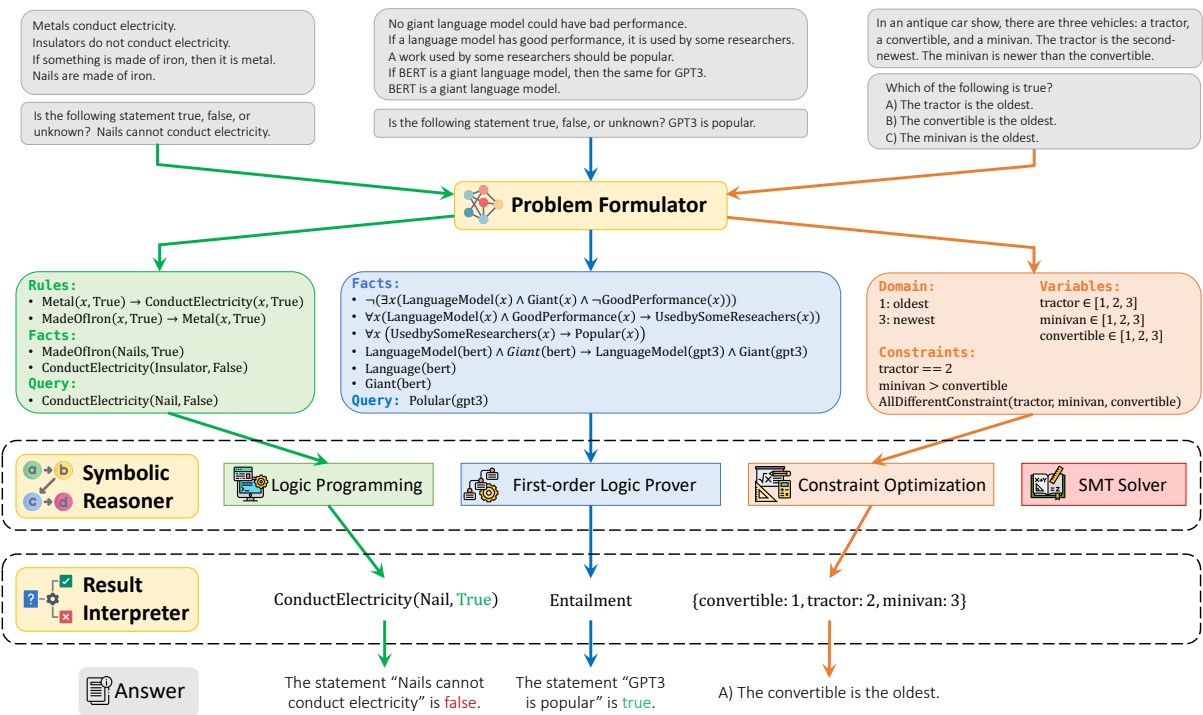

Figure 2: Overview of our LOGIC-LM model, which consists of three modules: (1) *Problem Formulator* generates a symbolic representation for the input problem with LLMs via in-context learning (2) *Symbolic Reasoner* performs logical inference on the formulated problem, and (3) *Result Interpreter* interprets the symbolic answer.

address this, recent work has begun to augment language models with access to external tools and resources, such as the information retriever (Nakano et al., 2021; Shi et al., 2023; Lazaridou et al., 2022), calculator (Cobbe et al., 2021), code interpreter (Wang et al., 2022), planner (Liu et al., 2023a), and other pre-trained models (Shen et al., 2023). Recent works (Gao et al., 2023; Chen et al., 2022) have achieved improved performance on arithmetic reasoning tasks by generating Python programs that specify the reasoning procedure as chained commands in the order of execution. However, this idea has not been extended to logical reasoning problems, primarily due to the challenge of representing their highly "non-linear" reasoning procedure (*e.g.*, hypothesizing, case-by-case analysis, and the process of elimination) with functional programming. Our work provides a novel way to solve this within the framework of augmented LLMs. Instead of parsing the problem-solving procedure as programs, we only describe the problem with symbolic language using LLMs and then offload the reasoning to external symbolic solvers.

**Auto-Formalization.** The concept of converting natural language into symbolic representations has been widely adopted in auto-formalization for mathematical reasoning (Wu et al., 2022; Drori

et al., 2022; He-Yueya et al., 2023; Jiang et al., 2023). These works demonstrate the proficiency of LLMs in translating a considerable fraction of mathematical problems into formal specifications defined in tools like SymPy (Meurer et al., 2017), Isabelle/HOL (Paulson, 1994), and Lean (de Moura et al., 2015). Mathematical reasoning can be considered a specialized subset of logical reasoning, primarily focused on numeric deductions. Due to this numeric specificity, mathematical problems are often more readily translatable to symbolic forms. In contrast, logical reasoning covers a wider array of problem types, often requiring a deeper understanding of world knowledge and commonsense for effective parsing into symbolic forms. Despite plenty of works studying mathematical reasoning, our work pioneers in extending the concept of auto-formalization to a broader range of logical reasoning tasks with modern LLMs.

## 3 LOGIC-LM

As shown in Figure 2, the inputs of our model are a logical reasoning problem $P$ described in natural language, along with a goal $G$ in the form of a multiple-choice or free-form question. LOGIC-LM then follows a *problem formulation-and-reasoning* paradigm to solve the problem.

In the *Problem Formulation* stage, we prompt an LLM to translate the problem and the goal into a task-specific symbolic language. In the *Symbolic Reasoning* stage, we call a deterministic symbolic solver, *e.g.*, a logic programming engine, to obtain a symbolic-represented answer. Finally, an LLM- or rule-based *Result Interpreter* is responsible for translating the answer back to natural language. Using this approach, the reasoning is guaranteed to be faithful as long as the problem formulation is correct since the answer $A$ is the result of executing deterministic algorithms (*e.g.*, forward/backward-chaining) embedded within the symbolic reasoner. Compared to previous methods based on chain-of-thought, our framework reduces the burden of LLMs by shifting their focus from "**solving** the problem by reasoning step-by-step" to "**representing** the problem in symbolic language".

## 3.1 Problem Formulator

Intuitively, LLMs may struggle with directly solving complex reasoning problems. However, they have demonstrated a notable ability to comprehend textual inputs and translate them into formal programs, such as mathematical equations (He-Yueya et al., 2023) or Python codes (Gao et al., 2023). We posit that this capability to *formulate problems into different languages* can be extended to symbolic languages as well. We leverage the few-shot generalization ability of LLMs to achieve this. By providing the LLM with detailed instructions about the grammar of the symbolic language, alongside a few demonstrations as in-context examples, we observe that LLMs, like InstructGPT (Ouyang et al., 2022b) and GPT-4 (OpenAI, 2023), can effectively follow the instructions to identify key entities, facts, and rules present in the problem statement, and then translate these elements into symbolic language following our defined grammar.

Specifically, we use four different symbolic formulations to cover four common types of logical reasoning problems: *deductive reasoning*, *first-order logic reasoning*, *constraint satisfaction problem*, and *analytical reasoning*. These formulations provide a foundation for translating natural language-based problem statements. By defining additional problem-specific formulations, our framework retains the flexibility to accommodate a wider range of reasoning tasks. Next, we will delve into the grammar of each symbolic formulation. Examples of each problem type are in Figure 2.

**Logic Programming (LP) Language.** Deductive reasoning typically starts from known facts and rules, and iteratively makes new inferences until the goal statement can be proved or disproved (Poole and Mackworth, 2010). The `Prolog` logic programming language (Clocksin and Mellish, 2003; Körner et al., 2022) is arguably the most prominent symbolic language to describe deductive reasoning problems. We adopt its grammar to represent a problem as facts, rules, and queries.

● **Facts**: a fact $F$ is a simple statement with a *predicate* and a set of *arguments*, formulated as $P(a_1, \cdots, a_n)$, where $P$ is the predicate name and each argument $a_i$ can be a variable, entity, number, or bool. For example, `Age(Peter, 31)` means "Peter's age is 31", and `MadeOfIron(Nails, True)` represents the fact "Nails are made of iron".

● **Rules**: rules are written in the form of clauses: $F_1 \wedge \cdots \wedge F_m \to F_{m+1} \wedge \cdots \wedge F_n$, where each $F_i$ is a fact and the rule means "if the facts $F_1, \cdots, F_m$ are true, then the facts $F_{m+1} \cdots F_n$ are also true."

● **Queries**: a query $Q$ is simply another fact required to be proved based on known facts and rules.

**First-Order Logic (FOL).** While the logic programming language efficiently represents common deductive reasoning problems, it may fail to represent more complex first-order logic (FOL) problems. To address this, we also include the FOL grammar (Enderton, 2001) in Appendix A. A problem is then parsed into a list of FOL formulas, which are divided into *Premises* (the known information from the problem) and *Conclusion* (the unknown formula to be proved). An example sentence and its FOL formula are given in Table 1.

**Constraint Satisfaction (CSP).** Constraint satisfaction problems (CSPs) (Kumar, 1992) aims to find the value assignment of a set of objects that satisfy a number of constraints. A CSP is often defined as a triple $(X, D, C)$, where $X = \{x_1, \cdots, x_n\}$ is a set of variables, $D = \{D_1, \cdots, D_n\}$ is a set of their respective domains of values, and $C = \{C_1, \cdots, C_m\}$ is a set of constraints. Each variable $x_i$ can take on the values in the nonempty domain $D_i$. Every constraint $C_j$ is a pair $\langle t_j, R_j \rangle$, where $t_j \subset X$ is a subset of $k$ variables and $R_j$ is a $k$-ary relation on the corresponding subset of domains $D_j$. We use the above syntax to define a CSP problem as variables, domains, and constraints. An example is given in both Figure 2 and Table 1.

| **Problem** | **Formulation** | **Example** | | **Solver** | **Dataset** |
| | | NL Sentence | Symbolic Formulation | | |
| --- | --- | --- | --- | --- | --- |
| Deductive Reasoning | LP | If the circuit is complete and the circuit has the light bulb then the light bulb is glowing. | `Complete(Circuit, True)∧` `Has(Circuit, LightBulb)` `→ Glowing(LightBulb, True)` | Pyke | ProntoQA, ProofWriter |
| First-Order Logic | FOL | A Czech person wrote a book in 1946. | $\exists x_2 \exists x_1 (\text{Czech}(x_1) \wedge \text{Author}(x_2, x_1)$ $\wedge \text{Book}(x_2) \wedge \text{Publish}(x_2, 1946))$ | Prover9 | FOLIO |
| Constraint Satisfaction | CSP | On a shelf, there are five books. The blue book is to the right of the yellow book. | `blue_book` $\in \{1,2,3,4,5\}$ `yellow_book` $\in \{1,2,3,4,5\}$ `blue_book > yellow_book` | python-constraint | LogicalDeduction |
| Analytical Reasoning | SAT | Xena and exactly three other technicians repair radios | `repairs(Xena, radios) ∧` `Count([t:technicians], t` $\neq$ `Xena` `∧ repairs(t, radios))) == 3)` | Z3 | AR-LSAT |

Table 1: A summary of the symbolic formulations (with examples) and symbolic solvers we use for the five datasets in our study, representing four different types of logical reasoning problems.

**Boolean Satisfiability (SAT) Formulation.** SAT is the problem of deciding if there is an assignment to the variables of a Boolean formula such that the formula is satisfied. Many analytical reasoning problems can be formulated as SAT problems. We adopt the grammar defined in Ye et al. (2023) to formulate an SAT problem $\mathcal{P}$ as $(\Phi, \mathcal{T}, \mathcal{Q})$, where $\Phi$ is a set of constraints defined under the theory $\mathcal{T}$, and $\mathcal{Q}$ is the query of interest.

Table 1 summarizes the four types of logical reasoning problems, their typical datasets, and the symbolic formulation used to represent each type of problem. We also give an example of a natural language statement with its corresponding symbolic formulation for each type. Appendix C shows the full prompts we use for the problem formulator. To teach LLMs to better align each statement with its corresponding symbolic form, we use the format SYMBOLIC_FORMULA ::: NL_STATEMENT in in-context examples to enable better grounding.

### 3.2 Symbolic Reasoner

After the problem formulator parses the problem $P$ and the goal $G$ into symbolic representations $\hat{P}$ and $\hat{G}$, we call a deterministic external solver depending on the task, to obtain the answer $A$. Table 1 summarizes the symbolic solvers we use for each type of logical reasoning problem.

**LP System.** For deductive reasoning, we incorporate the Pyke expert system (Frederiksen, 2008), which makes inferences based on the logic programming language. In response to a query, Pyke first creates a knowledge base, populating it with known facts and rules. Subsequently, it applies forward- and backward-chaining algorithms to infer new facts and substantiate the goal.

**FOL Prover.** We use Prover9[2] as the FOL inference engine. Prover9 is an automated theorem prover that supports first-order logic and equational logic. It initially converts FOL statements to conjunctive normal form (CNF) and then performs resolution (Robinson, 1965) on the CNF to deduce whether a conclusion is true, false, or unknown.

**CSP Solver.** Solving a CSP is to find value assignments for all variables that satisfy all given constraints. Commonly used algorithms for this task include backtracking, constraint propagation, and local search variants. To this end, we incorporate the python-constraint[3] package which offers solvers for CSPs over finite domains.

**SAT Solver.** For solving SAT problems, we use the Z3 theorem prover (de Moura and Bjørner, 2008), a satisfiability modulo theories (SMT) solver developed by Microsoft[4]. The SMT solver provides algorithms to determine whether a set of mathematical formulas is satisfiable. It generalizes the SAT problems to more complex formulas involving real numbers, integers, and various data structures such as lists, arrays, bit vectors, and strings. A lot of real-world analytical reasoning problems can be represented as problems of solving a system of equations.

### 3.3 Self-Refiner

For complex problems, generating the correct logical form may become challenging for LLMs. To address this, we introduce a *self-refinement* module that learns to modify inaccurate logical for-

---

[2] https://www.cs.unm.edu/~mccune/prover9/
[3] https://github.com/python-constraint/python-constraint
[4] https://github.com/Z3Prover/z3

mulations using the error messages from the symbolic reasoner as feedback. Recent works (Chen et al., 2023; Madaan et al., 2023) have adopted similar ideas to improve code generation, by teaching LLMs to debug their predicted programs via few-shot demonstrations. Here we extend this idea to refine generated logic representations. If the symbolic solver returns an execution error, we instruct the LLM to refine the incorrect logical form, by prompting it with the erroneous logic form, the solver's error message, and a set of demonstrations showing common error cases (*e.g.*, a free variable is not bounded to any quantifier in FOL) and their remedies. We run this process iteratively until either no error messages are returned, or the maximum number of allowable revisions is reached.

## 3.4 Result Interpreter

Finally, the result interpreter translates the results returned from the symbolic solver back to a natural language answer. For certain problems, this can be achieved through predefined rules; for example, mapping `Entailment` to `true`. However, this process can be more complex for CSPs, *e.g.*, translating *{convertible: 1, tractor: 2, minivan: 3}* to "*the convertible is the oldest.*". To handle these varying levels of complexity, we designed both rule-based and LLM-based result interpreters. Details of the result interpreter are given in Appendix D.

## 4 Experiments

**Datasets.** We evaluate LOGIC-LM on five common logical reasoning datasets, as follows.

**PrOntoQA** (Saparov and He, 2023) is a recent synthetic dataset created to analyze the capacity of LLMs for deductive reasoning. We use the hardest *fictional characters* version of the dataset, based on the results in Saparov and He (2023). Each version is divided into different subsets depending on the number of reasoning hops required. We use the hardest 5-hop subset for evaluation. Each question in PrOntoQA aims to validate a new fact's veracity, such as "True or false: Alex is not shy.".

**ProofWriter** (Tafjord et al., 2021) is another commonly used dataset for deductive logical reasoning. Compared with PrOntoQA, the problems are expressed in a more naturalistic language form. We use the open-world assumption (OWA) subset in which each example is a (problem, goal) pair and the label is one of {*PROVED, DISPROVED, UNKNOWN*}. The dataset is divided into five parts,

each part requiring $0, \leq 1, \leq 2, \leq 3$, and $\leq 5$ hops of reasoning, respectively. We evaluate the hardest *depth-5* subset. To reduce overall experimentation costs, we randomly sample 600 examples in the test set and ensure a balanced label distribution.

**FOLIO** (Han et al., 2022) is a challenging expert-written dataset for logical reasoning. The problems are mostly aligned with real-world knowledge and use highly natural wordings, and the questions require complex first-order logic reasoning to solve. We use the entire FOLIO test set for evaluation, consisting of 204 examples.

**LogicalDeduction** is a challenging logical reasoning task from the BigBench (Srivastava et al., 2022) collaborative benchmark. The problems are mostly about deducing the order of a sequence of objects from a minimal set of conditions. We use the full test set consisting of 300 examples.

**AR-LSAT** (Zhong et al., 2022) is a dataset that collects all analytical logic reasoning questions from the Law School Admission Test from 1991 to 2016. We use the test set which has 231 multiple-choice questions. AR-LSAT is particularly challenging, with state-of-the-art models only achieving performance slightly better than random guessing (Liang et al., 2022; Ribeiro et al., 2023a).

We convert all examples into a standard multiple-choice format, comprising a problem statement, a question, and potential answers, as shown in Figure 2. We also select 1-5 examples from the training set of each dataset as in-context examples. Detailed data statistics are in Appendix B.

**Baselines.** We compare our model against two baselines that depend solely on LLMs for logical reasoning: 1) *Standard* LLMs, which leverage in-context learning to directly answer the question; and 2) *Chain-of-Thought* (CoT) (Wei et al., 2022b), which adopts a step-by-step problem-solving approach, generating explanations before providing the final answer. We separately evaluate the settings that ChatGPT (`gpt-3.5-turbo`), GPT-3.5 (`text-davinci-003`) (Ouyang et al., 2022a) and GPT-4 (`gpt-4`) (OpenAI, 2023) serve as the underlying LLMs for all models. To ensure fair comparisons, we use the same in-context examples for all models. For reproducible results, we set the temperature to 0 and select the response with the highest probability from LLMs. Since all examples are formed as multiple-choice questions, we evaluate model performance based on the accuracy of selecting the correct answer.

| Dataset | ChatGPT (gpt-3.5-turbo) | | | GPT-3.5 (text-davinci-003) | | | GPT-4 (gpt-4) | | |
|---|---|---|---|---|---|---|---|---|---|
| | Standard | CoT | Logic-LM | Standard | CoT | Logic-LM | Standard | CoT | Logic-LM |
| PrOntoQA | 47.40 | **67.80** | 61.00 | 51.80 | 83.00 | **85.00** | 77.40 | **98.79** | 83.20 |
| ProofWriter | 35.50 | 49.17 | **58.33** | 36.16 | 48.33 | **71.45** | 52.67 | 68.11 | **79.66** |
| FOLIO | 45.09 | 57.35 | **62.74** | 54.60 | 57.84 | **61.27** | 69.11 | 70.58 | **78.92** |
| LogicalDeduction | 40.00 | 42.33 | **65.67** | 41.33 | 48.33 | **62.00** | 71.33 | 75.25 | **87.63** |
| AR-LSAT | 20.34 | 17.31 | **26.41** | 22.51 | 22.51 | **25.54** | 33.33 | 35.06 | **43.04** |

Table 2: Accuracy of standard promoting (Standard), chain-of-thought promoting (CoT), and our method (LOGIC-LM, without self-refinement) on five reasoning datasets. The best results within each base LLM are highlighted.

## 4.1 Main Results

We report the results of LOGIC-LM (*without* self-refinement) and baselines in Table 2. For LOGIC-LM, a symbolic solver does not return an answer when there are grammar errors in the symbolic formulation. For these un-executable cases, we fall back on using chain-of-thought to predict the answer. We have three major observations.

1. Logic-LM significantly outperforms standard LLMs and CoT across all datasets. With GPT-3.5, our method outperforms standard LLM on all datasets, with an average improvement of 39.2%. This highlights the benefit of combining LLMs with external symbolic solvers for logical reasoning. LOGIC-LM also improves CoT by a large margin of 18.4% on average, showing that offloading the reasoning to symbolic solvers greatly improves faithfulness compared with pure language-based reasoning with CoT.

2. GPT-4 outperforms GPT-3.5 by a large margin of 48.46% on average for the standard prompting. This aligns with the assertion that the main enhancement of GPT-4 lies in its ability to carry out complex reasoning (OpenAI, 2023). Although this may indicate that the logical reasoning capability can be boosted by scaling up the LLM, we observe that GPT-4 still makes numerous unfaithful reasoning errors. By delegating the reasoning to symbolic solvers, our method can further improve GPT-4 by an average of 24.98% and 10.44% for standard prompting and CoT prompting, respectively.

3. While integrating CoT generally enhances LLM performance, we find its benefits comparatively less substantial or even negative on FOLIO, LogicalDeduction, and AR-LSAT, with a modest improvement of 11.75%, 9.41%, and -3.2%, respectively. On the contrary, the benefits of CoT on PrOntoQA and ProofWriter are 51.59% and 33.82%, respectively. A plausible explanation is

| Dataset | SR | GPT-3.5 | | GPT-4 | |
|---|---|---|---|---|---|
| | | Exe_Rate | Exe_Acc | Exe_Rate | Exe_Acc |
| ProntoQA | − | 99.4% | 84.9 | 100.0% | 83.2 |
| | + | 100.0% ↑0.6 | 85.0 ↑0.1 | 100.0% | 83.2 |
| ProofWriter | − | 87.3% | 73.6 | 99.0% | 79.6 |
| | + | 95.6% ↑8.3 | 74.1 ↑0.5 | 99.0% | 79.6 |
| FOLIO | − | 66.7% | 61.8 | 79.9% | 80.4 |
| | + | 84.3% ↑17.6 | 64.3 ↑2.5 | 85.8% ↑5.9 | 79.9 ↓0.5 |
| Logical Deduction | − | 100.0% | 62.0 | 100.0% | 87.6 |
| | + | 100.0% | 62.0 | 100.0% | 87.6 |
| AR-LSAT | − | 11.3% | 57.7 | 32.6% | 60.0 |
| | + | 21.8% ↑10.5 | 60.3 ↑2.6 | 39.8% ↑7.2 | 58.8 ↓1.2 |

Table 3: Analysis of accuracy and execution status of LOGIC-LM. We present the percentage of executable logical formulations (Exe_Rate) together with the accuracy of the execution (Exe_Acc). SR represents before (−) and after (+) self-refinement.

that CoT emulates human forward-chain reasoning: beginning with known facts and sequentially deriving new conclusions until the goal is met. This reasoning style aligns well with problems in the PrOntoQA and ProofWriter datasets. However, FOL and CSP problems often necessitate more sophisticated reasoning strategies that are "non-linear" compared to standard forward-chain reasoning. These include hypothesizing, conditioning, recursive inference, and the process of elimination. Compared to CoT, the integration of symbolic solvers is better suited to these reasoning styles, hence yielding a more marked improvement on FOLIO (+21.85%), LogicalDeduction (+45.67%), and AR-LSAT (+24.14%).

## 4.2 Effectiveness of Problem Formulator

We then evaluate how well LLM can translate a given problem into the symbolic formulation used by each symbolic solver. In Table 3, we report the percentage of symbolic formulations that are *executable* by the corresponding symbolic solver for

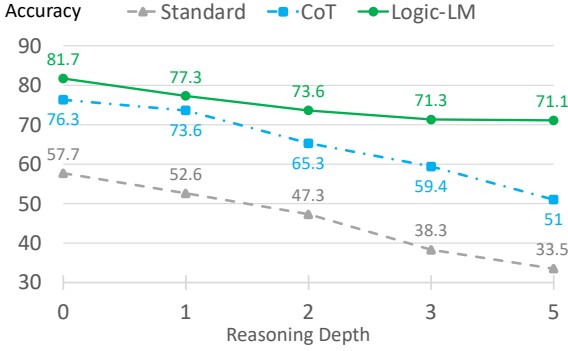

Figure 3: Accuracy of different models for increasing size of reasoning depth on the ProofWriter dataset.

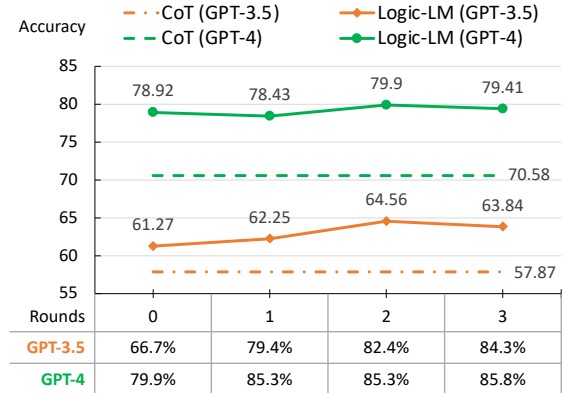

Figure 4: The accuracy for different rounds of self-refinement, with the corresponding executable rates.

each dataset (`Exe_Rate`). Generally, LLM demonstrates high proficiency in transcribing problems into symbolic formats, evidenced by its near 100% `Exe_Rate` on ProntoQA, ProofWriter, and LogicalDeduction. However, the high performance on these datasets is somewhat anticipated, given that their problems are mostly synthetically generated, limiting language variability. When it comes to datasets comprising real-world, expertly crafted problems, such as FOLIO and AR-LSAT, GPT-4's performance is notably less promising, with `Exe_Rate` scores of 79.9% and 32.6% respectively. This discrepancy underscores the inherent challenges associated with converting real-world problems into their logical equivalents.

`Exe_Rate` only reflects the *grammar* correctness of the logical form. We also report the accuracy of the executable samples (`Exe_Acc`) to measure the *semantic* correctness. We find that logical forms generated by GPT-4 generally achieve high `Exe_Acc`, even for the most challenging AR-LSAT dataset. Such performance accentuates the potential of symbolic solvers in bolstering the model's logical reasoning prowess, contingent on the precise translation of problems into symbolic forms.

### 4.3 Robustness of Reasoning

Incorporating symbolic solvers also leads to more *robust* reasoning. To illustrate this, we report the performance of LOGIC-LM and baselines for questions of varying complexity levels. We randomly selected 300 examples from each subset of ProofWriter, ensuring a balanced label distribution. The problems in these subsets require 0, <=1, <=2, <=3, and <=5 hops of reasoning, respectively. The results, shown in Figure 3, indicate that LOGIC-LM becomes increasingly effective as the required reasoning depth increases. For exam-

ple, LOGIC-LM outperforms CoT by 7.1%, 5.0%, 12.7%, 20.0%, and 39.4% on depth-0, depth-1, depth-2, depth-4, and depth-5 problems, respectively. In LOGIC-LM, multi-step logical reasoning is delegated to external symbolic solvers, thereby transitioning the challenge of LLM from *problem-solving* to *problem representation*. Ideally, the complexity of formally representing a problem statement in logical form should remain relatively constant, regardless of whether the questions require simple or complex reasoning. The trends in Figure 3 validate this assumption. The performance of *Standard* and *CoT* declines precipitously with the escalation of problem complexity. However, this trend is less prominent for LOGIC-LM, indicating that the robust reasoning capabilities provided by external solvers substantially mitigate performance degradation for complex reasoning problems.

### 4.4 Impact of Self-Refinement

In Table 3, we find that self-refinement is effective in fixing the in-executable symbolic formulations, increasing the `Exe_Rate` by 5.01 on average. For an in-depth analysis, we then evaluate the accuracy and `Exe_Rate` across different rounds of self-refinement on FOLIO, namely, 0 (no refinement), 1, 2, and 3 rounds. The results are in Figure 4.

We find that as the rounds of self-refinement increase, the percentage of executable formulations consistently increases, leading to an enhancement in the final performance. This suggests that self-refinement serves as an effective tool in aiding the LLM to accurately frame the problem. However, the accuracy tends to stagnate in subsequent rounds, even though the `Exe_Rate` continues to increase. This can be attributed to the type of feedback received by the self-refiner, which is the error mes-

```
Problem: "Stranger Things" is a popular Netflix show. If a Netflix show is popular, Karen will binge-watch it.
         If and only if Karen binge-watches a Netflix show, she will download it. Karen does not download "Black Mirror".
         "Black Mirror" is a Netflix show. If Karen binge-watches a Netflix show, she will share it to Lisa.
Question: Is the following statement true, false, or uncertain? "Black Mirror" is popular.
          (A) True              (B) False              (C) Uncertain
Generated symbolic Formulations:
   Predicates:
       NetflixShow(x) # x is a Netflix show.
       Popular(x) # x is popular.
       BingeWatch(x, y) # x binge-watches y.
       Download(x, y) # x downloads y.
       Share(x, y, z) # x shares y to z.
   Premises:
       NetflixShow(strangerThings) ∧ Popular(strangerThings) # "Stranger Things" is a popular Netflix show.
       ∀x (NetflixShow(x) ∧ Popular(x) → BingeWatch(karen, x)) # If a Netflix show is popular, Karen will binge-watch it.
       ∀x (NetflixShow(x) ∧ BingeWatch(karen, x) ↔ Download(karen, x)) # If and only if Karen … , she will download it.
       NetflixShow(blackMirror) ∧ ¬Download(karen, blackMirror) # Karen does not … "Black Mirror" is a Netflix show.
       ∀x (NetflixShow(x) ∧ BingeWatch(karen, x) → Share(karen, x, lisa)) # If Karen … , she will share it to Lisa.
   Conclusion:
       Popular(blackMirror) # "Black Mirror" is popular.
Predicted answer: B
```

Figure 5: An example of the generated symbolic representation and the predicted answer by LOGIC-LM.

sage from the symbolic solver. This feedback aids in converting "invalid" symbolic representations into valid ones. However, a valid symbolic representation does not necessarily equate to a "correct" problem formulation that accurately represents the problem. This issue could be tackled by enhancing the self-refiner to incorporate feedback beyond the error message, *e.g.*, a reward signal from an additional module evaluating the accuracy of a generated symbolic form. We leave this as a promising direction for future exploration.

### 4.5 Case Study

In Figure 5, we show an example of the symbolic representations generated by GPT-4, together with the predicted answer. In general, LOGIC-LM has demonstrated a potent capacity to interpret complex problems into symbolic forms. Nonetheless, there remain certain difficulties in accurately understanding the semantics of the problem.

We further analyze some error cases in Figure 6 of Appendix E. Example 1 shows a case where GPT-4 generates an incorrect FOL representation, stemming from its inability to define appropriate predicates. Here, instead of creating the predicate EasternWildTurkey, the model generates a constant, WildTurkey(eastern), in which WildTurkey is the predicate and eastern is the constant. While this representation is valid in isolation, it does not interact well with subsequent constants. This inconsistency is a recurring issue in GPT-4's symbolic form generation, illustrating that the model sometimes struggles to maintain an overarching understanding of the problem when forming logical symbols. Example 3 highlights a case where GPT-4 struggles to interpret specific

expressions accurately. In this case, the model fails to distinguish between the meanings of "below" and "above", resulting in an incorrect constraint Dan > Eve. Example 4 exemplifies GPT-4's challenge with fully grasping the rules of FOL grammar, evidenced by the invalid generated formula: Rating(subway, y) ∧ y > 9. These error cases underscore that transforming problems into logical forms remains a challenging task for modern LLMs, due to the intricacies of FOL formulation, the innate flexibility of natural language, and the complexity of global problem comprehension.

## 5 Conclusion and Future Work

In this work, we propose a novel approach to address logical reasoning problems by combining large language models with symbolic solvers. We introduce Logic-LM, one instantiation of such a framework, and demonstrate how it significantly improves performance over pure LLMs and chain-of-thought prompting techniques.

While Logic-LM has proven to be a capable system, it can be further improved with extension to more flexible and powerful logic systems. For example, statistical relational learning (SRL) systems such as Markov logic networks (Richardson and Domingos, 2006) and probabilistic soft logic (Bach et al., 2017) have demonstrated great promise in reasoning under uncertainty and integration with our framework would enable even more adaptive problem-solving capabilities. Additionally, our method can be extended to reasoning problems requiring commonsense, which remains a significant challenge as they often require reasoning over complex and ambiguous rules.

## Limitations

We identify two main limitations of LOGIC-LM. First, LOGIC-LM relies on translating reasoning problems into logical formats that can be tackled by symbolic solvers. As a consequence, the model's applicability is inherently bounded by the expressiveness of the symbolic solver, for example, not all problems can be easily encoded in first-order logic. Nevertheless, this limitation can be mitigated by integrating a more diverse set of symbolic solvers. The flexible design of LOGIC-LM facilitates this integration. The wide range of reasoning tasks that we can instantiate our LOGIC-LM framework on shows its general applicability.

Second, LOGIC-LM depends on in-context learning coupled with self-refinement to convert a natural language (NL) problem into the symbolic representation. While this method has proven to be effective, it may face difficulties when dealing with logical representations with intricate grammar structures, such as probabilistic soft logic. This arises from the difficulty in conveying complex grammatical rules to the language model through a limited number of demonstrations within a constrained context size. As a potential solution, future works could explore the development of specialized modules to enhance the mapping between NL and symbolic language, *e.g.*, fine-tuning LLMs with synthetic data generated via symbolic solvers.

## Ethics Statement

The use of large language models requires a significant amount of energy for computation for training, which contributes to global warming (Strubell et al., 2019). Our work performs few-shot in-context learning instead of training models from scratch, so the energy footprint of our work is less. The large language models whose API we use for inference, especially GPT-4, consume significant energy.

## Acknowledgements

This work was supported by the National Science Foundation Award #2048122. The views expressed are those of the authors and do not reflect the official policy or position of the US government.

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

## A Syntax for First-order Logic (FOL)

| Name | FOL Notation |
|------|--------------|
| Constant | lowercase letters |
| Variable | $x, y, z, \cdots$ |
| Atom | $P(a_1, \cdots, a_n)$ |
| Negation | $\neg P$ |
| Conjunction | $P_1 \wedge P_2$ 
 $P_1 \wedge, \cdots, \wedge P_n$ |
| Disjunction | $P_1 \vee P_2$ 
 $P_1 \vee, \cdots, \vee P_n$ |
| Implication | $P_1 \rightarrow P_2$ |
| Equivalence | $P_1 \leftrightarrow P_2$ |
| Existential Quantifier | $\exists x P(x, \cdots)$ |
| Universal Quantifier | $\forall x P(x, \cdots)$ |

Table 4: First-Order Logic Grammar.

## B Dataset Statistics

| Dataset | Reasoning | Test Size | #Opts |
|---------|-----------|-----------|-------|
| PrOntoQA | Deductive | 500 | 2 |
| ProofWriter | Deductive | 600 | 3 |
| FOLIO | FOL | 204 | 3 |
| LogicalDeduction | CSP | 300 | 3,5,7 |
| AR-LSAT | AR | 230 | 5 |

Table 5: Statistics of the logical reasoning datasets.

## C Prompt Examples

In this section we provide examples of the prompts used for each dataset and method. Prompts for standard in-context learning contain 2 demonstrations consisting of 3 parts each: a context, a question, and options. Prompts for chain-of-thought prompting contain 2 demonstrations consisting of 5 parts each: a task description, a context, a question, options, and a chain of reasoning. Prompts for Logic-LM contain 2 demonstrations with 5 parts each: a task description, a context, a question, options, and a domain-specific symbolic program. For brevity, *we show only a single demonstration* for each setting in the following sections.

### C.1 PrOntoQA Prompts

**Standard In-Context Learning**

```
Context: Jompuses are not shy. Jompuses are yumpuses.
(··· more context here ···)
Zumpuses are rompuses. Max is a yumpus.

Question: Is the following statement true or false?
Max is sour.

Options:
A) True
B) False

The correct option is: B
```

**Chain-of-Thought Prompting**

```
Task Description: Given a problem statement as
contexts, the task is to answer a logical reasoning
question.

Context: Jompuses are not shy. Jompuses are yumpuses.
(··· more context here ···)
Zumpuses are rompuses. Max is a yumpus.

Question: Is the following statement true or false?
Max is sour.

Options:
A) True
B) False

Reasoning: Max is a yumpus. Each yumpus is a dumpus.
(··· more reasoning here ···)
Tumpuses are not sour. So Max is not sour.

The correct option is: B
```

**Logic-LM**

```
Task Description: You are given a problem description
and a question. The task is to:
1) define all the predicates in the problem
2) parse the problem into logic rules based on
the defined predicates
3) write all the facts mentioned in the problem
4) parse the question into the logic form

Context: Each jompus is fruity.
(··· more context here ···)
Rompuses are zumpuses. Alex is a tumpus.

Question: True or false: Alex is not shy.

Predicates:
Jompus(\$x, bool) ::: Does x belong to Jompus?
(··· more predicates here ···)
Zumpus(\$x, bool) ::: Does x belong to Zumpus?

Facts:
Tumpuses(Alex, True)

Rules:
Jompus($x, True) >>> Fruity($x, True)
(··· more rules here ···)
Dumpus(\$x, True) >>> Rompus(\$x, True)

Query:
Shy(Alex, False)
```

## C.2 ProofWriter Prompts

### Standard In-Context Learning

```
Context: The cow is blue. The cow is round.
(··· more context here ···)
If the cow is cold and the cow visits the lion then
the lion sees the squirrel.

Question: Based on the above information, is the
following statement true, false, or unknown?
The tiger is not young.

Options:
A) True
B) False
C) Unknown

The correct option is: B
```

### Chain-of-Thought Prompting

```
Task Description: Given a problem statement as
contexts, the task is to answer a logical reasoning
question.

Context: The cow is blue. The cow is round.
(··· more context here ···)
If the cow is cold and the cow visits the lion then
the lion sees the squirrel.

Question: Based on the above information, is the
following statement true, false, or unknown?
The tiger is not young.

Options:
A) True
B) False
C) Unknown

Reasoning: The tiger likes the cow.
The tiger likes the squirrel.
(··· more reasoning here ···)
If something is nice and it sees the tiger then
it is young. So the tiger is young.

The correct option is: B
```

### Logic-LM

```
Task Description: You are given a problem description
and a question. The task is to:
1) define all the predicates in the problem
2) parse the problem into logic rules based on
the defined predicates
3) write all the facts mentioned in the problem
4) parse the question into the logic form

Context: Anne is quiet. Erin is furry.
(··· more context here ···)
All red people are young.

Question: Based on the above information, is the
following statement true, false, or unknown?
Anne is white.

Predicates:
Quiet($x, bool) ::: Is x quiet?
Furry($x, bool) ::: Is x furry?
(··· more predicates here ···)
White($x, bool) ::: Is x white?
Young($x, bool) ::: Is x young?

Facts:
Quite(Anne, True) ::: Anne is quiet.
(··· more facts here ···)
White(Harry, True) ::: Harry is white.

Rules:
Young($x, True) >>> Furry($x, True) ::: Young people
    are furry.
(··· more rules here ···)
Red($x, True) >>> Young($x, True) ::: All red people
    are young.

Query:
White(Anne, True) ::: Anne is white
```

## C.3 FOLIO Prompts

### Standard In-Context Learning

**Context**: All people who regularly drink coffee are dependent on caffeine.
(··· more context here ···)
If Rina is not a person dependent on caffeine and a student, then Rina is either a person dependent on caffeine and a student, or neither a person dependent on caffeine nor a student.

**Question**: Based on the above information, is the following statement true, false, or uncertain? Rina is a person who jokes about being addicted to caffeine or unaware that caffeine is a drug.

**Options**:
A) True
B) False
C) Uncertain

The correct option is: A

### Chain-of-Thought Prompting

**Task Description**: Given a problem statement as contexts, the task is to answer a logical reasoning question.

**Context**: The Blake McFall Company Building is a commercial warehouse listed on the National Register of Historic Places.
(··· more context here ···)
John works at the Emmet Building.

**Question**: Based on the above information, is the following statement true, false, or uncertain? The Blake McFall Company Building is located in Portland, Oregon.

**Options**:
A) True
B) False
C) Uncertain

**Reasoning**: The Blake McFall Company Building is another name for the Emmet Building.
(··· more reasoning here ···)
Therefore, the Blake McFall Company Building is located in Portland, Oregon.

The correct option is: A

### Logic-LM

**Task Description**: Given a problem description and a question. The task is to parse the problem and the question into first-order logic formulas. The grammar of the first-order logic formula is defined as follows:
1) logical conjunction: expr1 ∧ expr2
2) logical disjunction: expr1 ∨ expr2
3) logical exclusive disjunction: expr1 ⊕ expr2
4) logical negation: ¬expr1
5) expr1 implies expr2: expr1 → expr2
6) expr1 if and only if expr2: expr1 ↔ expr2
7) logical universal quantification: ∀ x
8) logical existential quantification: ∃ x
Output format: logic form ::: description

**Context**: All people who regularly drink coffee are dependent on caffeine.
(··· more context here ···)
If Rina is not a person dependent on caffeine and a student, then Rina is either a person dependent on caffeine and a student, or neither a person dependent on caffeine nor a student.

**Question**: Based on the above information, is the following statement true, false, or uncertain? Rina is either a person who jokes about being addicted to caffeine or is unaware that caffeine is a drug.

**Predicates**:
Dependent(x) ::: x is a person dependent on caffeine
(··· more predicates here ···)
Student(x) ::: x is a student

**Premises**:
∀x (Drinks(x) → Dependent(x)) ::: All people who regularly drink coffee are dependent on caffeine.
(··· more premises here ···)
∀x (Jokes(x) → ¬Unaware(x)) ::: No one who jokes about being addicted to caffeine is unaware that caffeine is a drug.

**Conclusion**:
Jokes(rina) ⊕ Unaware(rina) ::: Rina is either a person who jokes about being addicted to caffeine or is unaware that caffeine is a drug.

## C.4 LogicalDeduction Prompts

### Standard In-Context Learning

**Context**: The following paragraphs each describe a set of seven objects arranged in a fixed order.
(··· more context here ···)
Eve finished below Ada. Rob finished below Joe.

**Question**: Which of the following is true?

**Options**:
A) Ana finished third.
B) Eve finished third.
C) Ada finished third.
D) Dan finished third.
E) Rob finished third.
F) Amy finished third.
G) Joe finished third.

The correct option is: A

### Chain-of-Thought Prompting

**Task Description**: Given a problem statement as contexts, the task is to answer a logical reasoning question.

**Context**: The following paragraphs each describe a set of five objects arranged in a fixed order.
(··· more context here ···)
The raven is the third from the left.

**Question**: Which of the following is true?

**Options**:
A) The quail is the rightmost.
B) The owl is the rightmost.
C) The raven is the rightmost.
D) The falcon is the rightmost.
E) The robin is the rightmost.

**Reasoning**: The owl is the leftmost. This means owl is not the rightmost.
(··· more reasoning here ···)
This means raven is also not the rightmost. So, the answer is: A) The quail is the rightmost.

The correct option is: A

### Logic-LM

**Task Description**: You are given a problem description. The task is to parse the problem as a constraint satisfaction problem, defining the domain, variables, and contraints.

**Context**: The following paragraphs each describe a set of three objects arranged in a fixed order.
(··· more context here ···)
The minivan is newer than the convertible.

**Question**: Which of the following is true?

**Options**:
A) The station wagon is the second-newest.
B) The convertible is the second-newest.
C) The minivan is the second-newest.

**Domain**:
1: oldest
3: newest

**Variables**:
station\_wagon [IN] [1, 2, 3]
convertible [IN] [1, 2, 3]
minivan [IN] [1, 2, 3]

**Constraints**:
station\_wagon == 1 ::: The station wagon is the
    oldest.
minivan > convertible ::: The minivan is newer than
    the convertible.
AllDifferentConstraint([station\_wagon, convertible,
     minivan]) ::: All vehicles have different
    values.

**Query**:
A) station\_wagon == 2 ::: The station wagon is the
    second-newest.
B) convertible == 2 ::: The convertible is the
    second-newest.
C) minivan == 2 ::: The minivan is the second-newest
    .

## C.5 AR-LSAT Prompts

### Standard In-Context Learning

```
Context: During a single week, from Monday through
    Friday, tours will be conducted of a company's
    three divisions: Operations, Production, and
    Sales. Exactly five tours will be conducted
    that week, one each day. (··· more context here
    ···) If the Operations division is toured on
    Thursday, then the Production division is
    toured on Friday.

Question: Which one of the following CANNOT be true
    of the week's tour schedule?

Options:
A) The division that is toured on Monday is also
    toured on Tuesday.
B) The division that is toured on Monday is also
    toured on Friday.
C) The division that is toured on Tuesday is also
    toured on Thursday.
D) The division that is toured on Wednesday is also
    toured on Friday.
E) The division that is toured on Thursday is also
    toured on Friday.

The correct option is: C
```

### Chain-of-Thought Prompting

```
Task Description: Given a problem statement as
contexts, the task is to answer a logical reasoning
question.

Context: During a single week, from Monday through
    Friday, tours will be conducted of a company's
    three divisions: Operations, Production, and
    Sales. Exactly five tours will be conducted
    that week, one each day. (··· more context here
    ···) If the Operations division is toured on
    Thursday, then the Production division is
    toured on Friday.

Question: Which one of the following CANNOT be true
    of the week's tour schedule?

Options:
A) The division that is toured on Monday is also
    toured on Tuesday.
B) The division that is toured on Monday is also
    toured on Friday.
C) The division that is toured on Tuesday is also
    toured on Thursday.
D) The division that is toured on Wednesday is also
    toured on Friday.
E) The division that is toured on Thursday is also
    toured on Friday.

Reasoning: Since Thursday and Friday already have
tours planned, only Monday, Tuesday and Wednesday
tours need to be determined.
(··· more reasoning here ···)
A different division is toured on Thursday.
Therefore, the final answer is C.

The correct option is: C
```

### Logic-LM

```
Task Description: You are given a problem description.
The task is to parse the problem as a constraint
satisfaction problem, defining the domain,
variables, and contraints.

Context: A travel magazine has hired six interns -
    Farber, Gombarick, Hall, Jackson, Kanze, and
    Lha - to assist in covering three stories:
    Romania, Spain, and Tuscany. (··· more context here
    ···) Jackson is assigned to Tuscany. Kanze is
    not assigned to Spain.

Question: Which one of the following interns CANNOT
    be assigned to Tuscany?

Options:
(A) Farber
(B) Gombarick
(C) Hall
(D) Kanze
(E) Lha

Declarations:
stories = EnumSort([Romania, Spain, Tuscany])
assistants = EnumSort([photographer, writer])
(··· more declarations here ···)
trained = Function([interns] -> [assistants])

Constraints:
trained(Gombarick) == trained(Lha) ::: Gombarick and
    Lha will be trained in the same field
trained(Farber) != trained(Kanze) ::: Farber and
    Kanze will be trained in different fields
(··· more contraints here ···)
assigned(Jackson) == Tuscany ::: Jackson is assigned
    to Tuscany
assigned(Kanze) != Spain ::: Kanze is not assigned
    to Spain

Options:
is_unsat(assigned(Farber) == Tuscany) ::: (A)
is_unsat(assigned(Gombarick) == Tuscany) ::: (B)
is_unsat(assigned(Hall) == Tuscany) ::: (C)
is_unsat(assigned(Kanze) == Tuscany) ::: (D)
is_unsat(assigned(Lha) == Tuscany) ::: (E)
```

## D  Result Interpreter Implementation

For PrOntoQA and ProofWriter, the Pyke logic programming engine returns the inferred value of the variable in the query or Unknown if the variable cannot be determined. For example, for the query ConductElectricity(Nail, $x$), Pyke may return $x =$True. By comparing with the goal statement ConductElectricity(Nail, False), we can know that goal to be proved is False. For FOLIO, the FOL inference engine directly returns the veracity label of the goal as ENTAILMENT, CONTRADICTION, and CONTINGENT, which can be mapped to True, False, and Unknown, respectively. For LogicalDeduction, the solver returns all the possible value assignments in an array. We write rules to parse each option into the corresponding value and check it is in the generated array. For AR-LSAT, we attempt to separately prove each option to find the correct answer.

## E  Example Generations of LOGIC-LM

**Example 1**

**Problem:** There are six types of wild turkeys: Eastern wild turkey, Osceola wild turkey, Gould's wild turkey, Merriam's wild turkey, Rio Grande wild turkey, and Ocellated wild turkey. Tom is not an Eastern wild turkey. Tom is also not a Gould's wild turkey, or a Merriam's wild turkey, or a Rio Grande wild turkey. Tom is a wild turkey.
**Question:** Based on the above information, is the following statement true, false, or uncertain? Tom is an Ocellated wild turkey.
(A) True                    (B) False                    (C) Uncertain

**Predicted logic programs:**
    **Premises:**
    ~~WildTurkey(eastern) ∧ WildTurkey(osceola) ∧ WildTurkey(gould) ∧ WildTurkey(merriams) ∧ WildTurkey(rio_grande) ∧ WildTurkey(ocellated)~~
    → ∀x (WildTurkey(x) → (OcellatedWildTurkey(x) ∨ EasternWildTurkey(x) ∨ OsceolaWildTurkey(x) ∨ GouldsWildTurkey(x) ∨ MerriamsWildTurkey(x)
    ∨ RioGrandeWildTurkey(x)))
    WildTurkey(tom) ∧ ¬EasternWildTurkey(tom) ∧ ¬OsceolaWildTurkey(tom) ∧ ¬GouldsWildTurkey(tom) ∧ ¬MerriamsWildTurkey(tom) ∧
    ¬RioGrandeWildTurkey(tom)
    **Conclusion:**
    OcellatedWildTurkey(tom)
**Predicted answer:** ~~C~~ → A

**Example 2**

**Problem:** A Japanese game company created the game the Legend of Zelda. All games in the Top 10 list are made by Japanese game companies. If a game sells more than one million copies, then it will be selected into the Top 10 list. The Legend of Zelda sold more than one million copies.
**Question:** Based on the above information, is the following statement true, false, or uncertain? The Legend of Zelda is in the Top 10 list.
(A) True                    (B) False                    (C) Uncertain

**Predicted logic programs:**
    **Premises:**
    JapaneseGameCompany(legend_of_zelda)
    ∀x (Top10List(x) → JapaneseGameCompany(x))
    ∀x (SellMoreThanOneMillionCopies(x) → Top10List(x))
    SellMoreThanOneMillionCopies(legend_of_zelda)
    **Conclusion:**
    Top10List(legend_of_zelda)
**Predicted answer:** A

**Example 3**

**Problem:** The following paragraphs each describe a set of five objects arranged in a fixed order. The statements are logically consistent within each paragraph. In a golf tournament, there were five golfers: Rob, Eve, Eli, Amy, and Dan. Dan finished second. Amy finished below Eve. Dan finished above Eve. Amy finished above Eli.
**Question:** Which of the following is true?
(A) Rob finished third    (B) Eve finished third    (C) Eli finished third    (D) Amy finished third    (E) Dan finished third

**Predicted logic programs:**
    **Domain:**                **Variables:**                        **Constraints:**
    1: first                   Rob [IN] [1, 2, 3, 4, 5]               Dan == 2
    5: last                    Eve [IN] [1, 2, 3, 4, 5]               Amy > Eve
                               Eli [IN] [1, 2, 3, 4, 5]               ~~Dan > Eve~~ → Dan < Eve
                               Amy [IN] [1, 2, 3, 4, 5]               Amy < Eli
                               Dan [IN] [1, 2, 3, 4, 5]               AllDifferentConstraint([Rob, Eve, Eli, Amy, Dan])
    **Query:**
    (A) Rob == 3    (B) Eve == 3    (C) Eli == 3    (D) Amy == 3    (E) Dan == 3
**Predicted answer:** ~~E~~ → B

**Example 4**

**Problem:** If the restaurant is listed in Yelp's recommendations, then the restaurant does not receive many negative reviews. All restaurants with a rating greater than 9 are listed in Yelp's recommendations. Some restaurants that do not provide take-out service receive many negative reviews. All restaurants that are popular among local residents have ratings greater than 9. Subway has a rating greater than 9 or is popular among local residents.
**Question:** Based on the above information, is the following statement true, false, or uncertain? Subway provides take-out service and does not receive many negative reviews.
(A) True                    (B) False                    (C) Uncertain

**Predicted logic programs:**
    **Premises:**
    ∀x (Listed(x) → ¬NegativeReviews(x))
    ~~∀x (Rating(x, y) ∧ y > 9 → Listed(x))~~ → ∀x (GreaterThanNine(x) → Listed(x))
    ∃x (¬TakeOut(x) ∧ NegativeReviews(x))
    ~~∀x (Popular(x) → (Rating(x, y) ∧ y > 9))~~ → ∀x (Popular(x) → GreaterThanNine(x))
    ~~Rating(subway, y) ∧ y > 9 ∨ Popular(subway)~~ → GreaterThanNine(subway) ∨ Popular(subway)
    **Conclusion:**
    TakeOut(subway) ∧ ¬NegativeReviews(subway)
**Predicted answer:** ~~C~~ → A

Figure 6: Examples of generated symbolic representations and predicted answers. The incorrect segment(s) and their correspondence in the problem are marked in **red**, and the correct revisions are marked in **green**.