# OpenReview forum: "Logic-LM: Empowering Large Language Models with Symbolic Solvers for Faithful Logical Reasoning"
_EMNLP/2023/Conference — EMNLP 2023 Findings_

### Official Review · Reviewer_sDiS · 2023-07-29

**Soundness:** 3

**Excitement:**

3: Ambivalent: It has merits (e.g., it reports state-of-the-art results, the idea is nice), but there are key weaknesses (e.g., it describes incremental work), and it can significantly benefit from another round of revision. However, I won't object to accepting it if my co-reviewers champion it.

**Missing References:**

What the “Problem Formulator” module does is similar to the autoformalization task in mathematical reasoning, for instance:
[1] Wu Y, Jiang A Q, Li W, et al. Autoformalization with large language models[J]. Advances in Neural Information Processing Systems, 2022, 35: 32353-32368.
[2] Wang Q, Brown C, Kaliszyk C, et al. Exploration of neural machine translation in autoformalization of mathematics in Mizar[C]//Proceedings of the 9th ACM SIGPLAN International Conference on Certified Programs and Proofs. 2020: 85-98.


**Paper Topic And Main Contributions:**

This paper proposes integrating LLMs with symbolic solvers to improve logical problem-solving. A self-refiner is incorporated in the reasoning process to iteratively refine the generate valid logic forms. The final answer is then obtained by a result interpreter. The method is evaluated on 4 datasets.

**Questions For The Authors:**

Question A: The framework integrates three sets of grammar: deductive reasoning, first-order logic, and constraint optimization. In practice, how to choose and designate from the three grammars for a specific reasoning question?
Question B: How does this framework deals with error propagation?


**Reasons To Accept:**

The experimental results show the effectiveness of the method. The paper is clearly written and easy to follow.

**Reasons To Reject:**

The idea of parsing natural language into symbolic forms might not be original considering the autoformalization task in mathematical reasoning. Please see the details in “Missing References”.

**Reproducibility:**

3: Could reproduce the results with some difficulty. The settings of parameters are underspecified or subjectively determined; the training/evaluation data are not widely available.

**Reviewer Confidence:**

4: Quite sure. I tried to check the important points carefully. It's unlikely, though conceivable, that I missed something that should affect my ratings.

---

> ### Author Rebuttal · Authors · 2023-08-27
>
> We appreciate the valuable insights behind the comments and also want to clarify a few points.
>
> > *The idea of parsing natural language into symbolic forms might not be original considering the autoformalization task in mathematical reasoning.*
>
> We appreciate the reviewer's insight regarding the similarities between Logic-LM and the autoformalization tasks common in mathematical reasoning. While the core idea of translating natural language into symbolic forms might be similar, the two areas differ in scope and complexity. Logical reasoning encompasses a broad scope of making and justifying arguments. As such, mathematical reasoning can be considered a specialized subset of logical reasoning, primarily focused on numeric deductions. Due to this numeric specificity, mathematical problems are often more readily translatable to symbolic forms. In contrast, logical reasoning covers a wider array of problem types, often requiring a deeper understanding of world knowledge and commonsense for effective parsing into symbolic forms.
>
> Despite plenty of works studying mathematical reasoning, our work pioneers in extending the concept of "autoformalization" to a broader range of logical reasoning tasks with LLMs. Nonetheless, we acknowledge the relevance of existing works on autoformalization in mathematical reasoning and will include the suggested references in the related work section of our paper.
>
> > *Question A: The framework integrates three sets of grammar: deductive reasoning, first-order logic, and constraint optimization. In practice, how to choose and designate from the three grammars for a specific reasoning question?*
>
> As an initial exploration of integrating symbolic solvers with LLMs, our work primary focus on how to correctly translate the problem into the specific symbolic formulation used by the chosen symbolic solver. Therefore, in our experiments, the grammar for each dataset was predetermined. However, we agree with the reviewer's point that automatically selecting an appropriate symbolic system is an important practical concern.
>
> **Additional Experiments**. To address this, we conducted an additional experiment to evaluate the potential of using LLMs as grammar designators. We prompted GPT-4 with 8 demonstrations, each consisting of a problem and its corresponding symbolic language, such as `(Problem from FOLIO, "FOL representation")`. These demonstrations were randomly drawn from each of the four datasets we used (two examples per dataset).  Subsequently, we evaluated GPT-4's performance on the grammar designation task to assign new examples to the appropriate symbolic grammar (3-class classification since we have defined three different grammars). Out of 400 randomly sampled test cases (100 from each dataset), GPT-4 achieved a high F1 score of 96.5%. However, we note that this high performance may partially due to the fact that the problems of the four datasets we used are easy to differentiate. In real-world applications, determining the most fitting symbolic system could be more challenging. We will include these new findings and discussions in a discussion section of the camera-ready version. We thank the reviewer for raising this interesting and valuable question.
>
> > *Question B: How does this framework deals with error propagation?*
>
> First, we want to clarify that Logic-LM actually has less error propagation problems than chain-of-thought.  CoT sequentially generates the reasoning chain step-by-step until it reaches the final answer. Therefore, error propagation is easy to happen where an error in early reasoning steps can readily affect all subsequent steps. Logic-LM mitigates this by refocusing the language model's task from "step-by-step problem-solving" to "symbolic problem representation" (Line 203 - 207). When the problem is accurately formulated, symbolic solvers can guarantee the correctness of the reasoning. By offloading the burden of logical reasoning to external symbolic solvers, it minimizes the risk of error propagation for CoT during complex reasoning processes.
>
> Second, we admit that Logic-LM is not entirely immune to error propagation. Errors can occur when the LLM generates incorrect symbolic formulations that affect the solver's reasoning. To address this, we have integrated the self-refinement mechanism that uses feedback from the symbolic solver's error messages to correct faulty formulations (Section 3.4). Self-refinement can effectively alleviate error propagation, as evidenced by the results in Table 3 and Figure 5. However, as mentioned in Lines 543-550, there still exist cases where incorrect formulations go undetected by the symbolic solver. To tackle this limitation, future work could involve extending the self-refinement module to incorporate additional forms of feedback, such as a reward signal from a module specifically designed to assess the accuracy of generated symbolic forms. We leave this for future exploration.

---

### Official Review · Reviewer_udfF · 2023-08-03

**Soundness:** 4

**Excitement:**

4: Strong: This paper deepens the understanding of some phenomenon or lowers the barriers to an existing research direction.

**Missing References:**

N/A

**Paper Topic And Main Contributions:**

This paper proposes a new two-stage framework addressing the task of performing logical reasoning from a set of premises: first, it converts the premises and conclusion into an intermediate first-order logical representation called Pyke. Then, it applies an inference engine (one from the Stanford CS221 course) to deduce the truthness of the resulting conclusion.

**Questions For The Authors:**

1. The benefit of Logic-LM over CoT is clear for the highest reasoning depths, but when reasoning depth is lower, to what extent does Logic-LM improve over CoT? Is it possible that the failure rate of the translation (as shown in Fig. 6) ends up being worse than the failure rate of CoT logical errors (as ther is only one step of reasoning)?

2. The four datasets you test on are still rather synthetic datasets. How do you envision your Logic-LM approach to apply in more complex and real-world applications?

**Reasons To Accept:**

1. One of the biggest failure modes of language models is their inability to perform faithful logical reasoning, and Logic-LM no longer requires the language model to perform the logical reasoning. In a sense, there is a clean separation of concerns here between the language model and the symbolic solver. This neurosymbolic approach to logical reasoning is likely to inspire future work that doesn't rely solely on the LM.

2. The symbolic solver in Logic-LM will never fault on performing logical reasoning. Errors when using the Logic-LM framework would be much easier to debug.

3. Table 2 shows reasonable gains by using Logic-LM compared to CoT and standard prompting on three different models across four different datasets, highlighting that the technique is widely applicable.



**Reasons To Reject:**

1. The first order logic representation used may be somewhat complex and difficult for someone to understand if they need to go in and debug the full pipeline, compared to the explanation generated by chain of thought.

2. This approach works well if the premises can be cleanly converted into FOL representations, but this leads to a rather restricted set of possible applications for the Logic-LM framework.

3. Sometimes, CoT performance is very similar to Logic-LM performance, highlighting that CoT is perhaps already able to perform some degree of logical reasoning. The work does provide a few examples of incorrect first-order logic representations, but does not go in depth to analyze if CoT failures are systematic and can be recovered by Logic-LM.

**Reproducibility:**

5: Could easily reproduce the results.

**Reviewer Confidence:**

4: Quite sure. I tried to check the important points carefully. It's unlikely, though conceivable, that I missed something that should affect my ratings.

**Typos Grammar Style And Presentation Improvements:**

- It might be of interest to include a brief idea of how the CS221 solver works
- Including a few examples of the benefit of self-refine would be instructive

---

> ### Author Rebuttal · Authors · 2023-08-27
>
> We appreciate the valuable insights behind the comments and also want to clarify a few points.
>
> > *The first order logic representation used may be somewhat complex and difficult for someone to understand if they need to go in and debug the full pipeline, compared to the explanation generated by chain of thought.*
>
> We appreciate the reviewer's concern regarding the explainability of our neuro-symbolic approach in Logic-LM. While it's true that methods like chain-of-thought offer better natural language explainability, our primary focus in this paper is on enhancing the **faithfulness** of LLMs in logical reasoning. The use of symbolic language, although less interpretable, provides a more precise and verifiable form of reasoning.
>
> To strike a balance between faithfulness and explainability, we have made some initial efforts. Specifically, we prompt Logic-LM to generate both the symbolic formula and its corresponding natural language explanation (as mentioned in Line 297). This not only helps in better grounding of the symbolic language but also offers users a level of explainability. Looking ahead, we see ample opportunity to further improve Logic-LM's explainability. One promising direction could involve introducing a *Reasoning Explanation* module designed to translate the symbolic solver's reasoning path into easily understandable natural language. We will add this discussion into the future work section. We appreciate the reviewer highlighting this important aspect.
>
> > *The four datasets you test on are still rather synthetic datasets. How do you envision your Logic-LM approach to apply in more complex and real-world applications?*
>
> Thanks for pointing this out. We agree with the reviewer's point that we should evaluate on more real-world datasets. Therefore, we present additional experimental results on two realistic human-created datasets: GSM8K and AR-LSAT.
>
> - GSM8K [1] contains 8.5K high-quality mathematical logic reasoning problems, crafted by human experts. For our evaluation, we use the test set, containing 1,319 questions.
> - AR-LSAT [2] collected all analytical logic reasoning questions from the Law School Admission Test from 1991 to 2016. We use the test set which has 231 multiple-choice questions.
>
> For GSM8K, we employ *SymPy*, a Python library specialized in symbolic mathematics, to serve as the symbolic reasoner. For AR-LSAT, we utilize the *Z3 theorem prover* as the symbolic reasoner. The results of Logic-LM (without self-refinement) and the baselines are given in the table below.
>
> |          |           | GSM8K     |           |           | AR-LSAT   |           |
> | -------- | --------- | --------- | --------- | --------- | --------- | --------- |
> |          | GPT-3.5   | ChatGPT   | GPT-4     | GPT-3.5   | ChatGPT   | GPT-4     |
> | Standard | 23.95     | 15.77     | 43.32     | 20.34     | 22.51     | 33.33     |
> | CoT      | 65.04     | 60.72     | 90.36     | 17.31     | 22.51     | 35.06     |
> | Logic-LM | **69.59** | **64.44** | **91.04** | **21.64** | **30.73** | **45.45** |
>
> We find that Logic-LM without self-refinement surpasses both Standard and CoT prompting on both datasets across all tested models, including GPT-3.5, ChatGPT, and GPT-4. This shows that Logic-LM can also be applied in more complex and real-world applications. The integration of a symbolic solver, such as Z3 in this case, considerably enhances the model's logical reasoning capabilities by translating real-world problems into symbolic language. We will add these additional results to the final version.
>
> - [1] Zhong et al., AR-LSAT: Investigating Analytical Reasoning of Text. NAACL, 2022.
> - [2] Cobbe et al., Training Verifiers to Solve Math Word Problems. 2021.
>
> > *This approach works well if the premises can be cleanly converted into FOL representations, but this leads to a rather restricted set of possible applications for the Logic-LM framework.*
>
> Thanks for pointing this out. First, we want to clarify that Logic-LM is not solely tied to FOL representations. Instead, it is a versatile framework that integrates LLMs with various symbolic solvers, and FOL is only one of the symbolic formulations. Together with the additional results shown in the previous reply, we already demonstrated the applicability of Logic-LM on six different logical reasoning datasets, and we have integrated a diverse set of symbolic solvers, including tools for symbolic mathematics (SymPy), theorem proving (Z3), first-order logic (FOL), constraint satisfaction problem (CSP solver), and logic programming (Pyke). This adaptability of a wide range of tasks and symbolic solvers underlines the general utility of Logic-LM.
>
> Second, we acknowledge in our limitations section that Logic-LM is confined by the expressiveness of the symbolic solvers it employs. While certain problems may not be represented as logical form, this limitation should not diminish the inherent value of integrating symbolic solvers with LLMs. Looking ahead, an intriguing direction for extending Logic-LM's applicability would be to explore a hybrid approach that combines language-based reasoning (e.g., CoT) with symbolic-based reasoning.
>
> > *The work does provide a few examples of incorrect first-order logic representations, but does not go in depth to analyze if CoT failures are systematic and can be recovered by Logic-LM.*
>
> We appreciate the reviewer's emphasis on the need for a more rigorous analysis of CoT's failure modes. In Section 4.1 (Lines 454 - 475), we have touched upon CoT's limitations in "non-linear" reasoning that include hypothesizing, conditioning, recursive inference, and the process of elimination. We argued that the integration of symbolic solvers is better suited to these reasoning styles, hence yielding a more marked improvement on the FOLIO (+15.67%) and the LogicalDeduction (+56.85%) datasets. In the camera-ready version, we plan to enrich this part with a more comprehensive quantitative analysis. This involves categorizing and reporting the proportion of each prevalent failure mode in CoT and discussing how Logic-LM effectively addresses these issues. We thank the reviewer for highlighting this point.
>
> > *The benefit of Logic-LM over CoT is clear for the highest reasoning depths, but when reasoning depth is lower, to what extent does Logic-LM improve over CoT? Is it possible that the failure rate of the translation (as shown in Fig. 6) ends up being worse than the failure rate of CoT logical errors (as ther is only one step of reasoning)?*
>
> As shown in Figure 3, when the reasoning depth is 0, the Logic-LM and CoT achieve 81.7 and 76.3 accuracy on ProofWriter, respectively. Although the benefit of Logic-LM is less obvious than that in higher reasoning depths, Logic-LM still achieved an improvement of 7.1%. As shown in Table 3, Logic-LM generally show a very low failure rate in symbolic translation. For problems with low reasoning depth, the failure rate is even lower. For example, in ProofWriter depth-1, we find that GPT-3.5 produces 100% executable programs, 77.3% of which yielded correct answers, outperforming CoT's 73.6% accuracy. Unfortunately, we could only do the analysis on ProofWriter because the other datasets do not annotate the reasoning depth for their problems.

---

### Official Review · Reviewer_Yhix · 2023-08-09

**Soundness:** 3

**Excitement:**

3: Ambivalent: It has merits (e.g., it reports state-of-the-art results, the idea is nice), but there are key weaknesses (e.g., it describes incremental work), and it can significantly benefit from another round of revision. However, I won't object to accepting it if my co-reviewers champion it.

**Paper Topic And Main Contributions:**

This paper takles logical reasoning problems by transfer four existing synthetic natural language datasets (where each data in the datasets is generated automatically by transfering symbolic reasoner generated symbolic sentences to synthetic natural language sentences) back to symbolic sentences and solve it with symbolic reasoner. This paper tries to indicate that in this situation using a symbolic reasoner can achieve better performance than using LM to directly solve the synthetic natural language tasks.

**Questions For The Authors:**

1. what is the performance of proposed Logic-LM on Entailmentbank dataset? How is it compared to "Standard" and "COT"?
2. what if "Standard" and "COT" use self-refinement to directly solve the original problems?
3. what can I learn from this paper given my previous analysis of this paper?

**Reasons To Accept:**

1. A through analysis on the performance of transfering synthetic natural language back to logic language and solve it with symbolic solver.

**Reasons To Reject:**

1. The adopted datasets are not persuading to be used by this paper to form its conclusions. Because every data in the four datasets is generated from symbolic reasoner and tranfered to synthetic natural language. The goal the the four datasets is to analyze the existing LMs' ability on logical reasoning on natural language. It is very improper to use these four datasets, and transfer their data back to symbolic language to be solved by symbolic reasoner to illustrate the power of symbolic reasoner. However, if the method of this paper can work on any dataset that is not automatically generated with symbolic reasoner, I probably would agree that it is a good work. An example is Entailmentbank [1]. However, I have not seen this paper working on similar datasets.
2. The performance comparision is established in an unfair experienment --- Logic-LM can use self-refinement to refine the transfered symbolic language, then why "Standard" and "COT" can't use self-refinement as well to directly solve the original problem? I don't think any number in table 2 is persuading.
3. It is hard to learn anything from this paper because of my previous reaons.


[1] Dalvi, B., Jansen, P., Tafjord, O., Xie, Z., Smith, H., Pipatanangkura, L., & Clark, P. (2021). Explaining answers with entailment trees. arXiv preprint arXiv:2104.08661.

**Reproducibility:**

4: Could mostly reproduce the results, but there may be some variation because of sample variance or minor variations in their interpretation of the protocol or method.

**Reviewer Confidence:**

4: Quite sure. I tried to check the important points carefully. It's unlikely, though conceivable, that I missed something that should affect my ratings.

---

> ### Author Rebuttal · Authors · 2023-08-26
>
> > *The adopted datasets are not persuading to be used by this paper to form its conclusions. Because every data in the four datasets is generated from symbolic reasoner and tranfered to synthetic natural language.*
>
> We want to clarify that FOLIO is not a dataset automatically generated with symbolic reasoner. FOLIO is an expert-written dataset for FOL reasoning equipped with parallel FOL formulas. The examples are aligned with real-world knowledge and use highly natural wordings. They asked the annotators to create new FOL problems from scratch, using Wikipedia articles to develop ideas for topics (please refer to Section 3.1 of the FOLIO paper). They also use a hybrid of automatic generation and manual annotation. In this case, only the problem template is automatically generated. Human annotators are asked to create problems based on the template. Therefore, FOLIO is clearly not a "synthetic dataset automatically generated with symbolic reasoner".
>
> ---
>
> > *However, if the method of this paper can work on any dataset that is not automatically generated with symbolic reasoner, I probably would agree that it is a good work.*
>
> Although we don't agree that all datasets used in this paper are automatically generated with symbolic reasoner, we appreciate the reviewer's point that evaluating on more real-world datasets could make our work more convincing. Therefore, we present additional experimental results on two realistic human-created datasets: GSM8K and AR-LSAT.
>
> **Datasets**
>
> - GSM8K [1] contains 8.5K high-quality mathematical logic reasoning problems, crafted by human experts. For our evaluation, we use the test set, containing 1,319 questions.
> - AR-LSAT [2] collected all analytical logic reasoning questions from the Law School Admission Test from 1991 to 2016. We use the test set which has 231 multiple-choice questions.
>
> **Symbolic Language and Reasoners.**  For GSM8K, we employ *SymPy*, a Python library specialized in symbolic mathematics, to serve as the symbolic reasoner. For AR-LSAT, we utilize the *Z3 theorem prover* as the symbolic reasoner. Below, we provide examples of questions from each dataset, along with their corresponding symbolic representations.
>
> **GSM8K: Problem**
>
> ```
> Question: Olivia has $23. She bought five bagels for $3 each. How much money does she have left?
> ```
>
> **GSM8K: symbolic formulation**
>
> ```python
> # Variables
> Money_of_Olivia = 23 # Olivia has $23
> Each_Bagel_Price = 3 # Each bagel is $3
> Num_Bagel_Olivia_Bought = 5 # Olivia bought five bagels
>
> # Query
> Money_of_Olivia - Each_Bagel_Price * Num_Bagel_Olivia_Bought # How much money does she have left?
> ```
>
> **AR-LSAT: Problem**
>
> ```
> As part of an open house at a crafts studio, three teachers—Jiang, Kudrow, and Lanning—will give six consecutive presentations on six different subjects. Jiang will present on needlework and origami; Kudrow on pottery, stenciling, and textile making; and Lanning on woodworking. The order of their presentations will meet the following conditions: Kudrow cannot give two presentations in a row. The presentation on stenciling must be given earlier than the one on origami. The presentation on textile making must be given earlier than the one on woodworking.
>
> If textile making is presented fifth, which one of the following could be true?
> A) Needlework is presented sixth.
> B) Pottery is presented fourth.
> C) Stenciling is presented second.
> D) Stenciling is presented third.
> E) Woodworking is presented second.
> ```
>
> **AR-LSAT: symbolic formulation**
>
> ```python
> # Variables and functions
> Subjects = [needlework, origami, pottery, stenciling, textile_making, woodworking]
> Teachers = [Jiang, Kudrow, Lanning]
> Position = Function(Subjects -> bool)
>
> # Constraints
> ForAll([sub IN Subjects], And(position(sub) >= 1, position(sub) <= 6)) # each position should be from 1 to 6
> ForAll([s_1, s_2], Implies(s_1 != s_2, position(s_1) != position(s_2))) # each subject should be assigned a different order
> teach(needlework) == Jiang # Jiang will present on needlework
> teach(origami) == Jiang # Jiang will present on origami
> teach(pottery) == Kudrow # Kudrow will present on pottery
> teach(stenciling) == Kudrow # Kudrow will present on stenciling
> teach(textile_making) == Kudrow # Kudrow will present on textile making
> teach(woodworking) == Lanning # Lanning will present on woodworking.
> position(textile_making) == 5 # textile making is presented fifth.
> position(stenciling) < position(origami) # The presentation on stenciling must be given earlier than the one on origami.
> position(textile_making) < position(woodworking) # The presentation on textile making must be given earlier than the one on woodworking.
> ForAll([s_1, s_2], Implies(And(teach(s_1) == Kudrow, teach(s_2) == Kudrow, s_1 != s_2), And(Not(position(s_1) == position(s_2) - 1), Not(position(s_2) == position(s_1) - 1)))) # Kudrow cannot give two presentations in a row.
>
> # Query
> position(needlework) == 6 # A) Needlework is presented sixth.
> position(pottery) == 4 # B) Pottery is presented fourth.
> position(stenciling) == 2 # Stenciling is presented second.
> position(stenciling) == 3 # Stenciling is presented third.
> position(woodworking) == 2 # Woodworking is presented second.
> ```
>
> **Main Results.** The results of Logic-LM (without self-refinement) and the baselines are given in the table below. Consistent with Table 3 in our paper, *Exe_Rate* means the percentage of executable symbolic representations, and *Exe_Acc* means the accuracy of the executable samples. We use the same set of in-context examples for Standard, CoT, and Logic-LM for fair comparison.
>
> |          |           | GSM8K     |           |           | AR-LSAT   |           |
> | -------- | --------- | --------- | --------- | --------- | --------- | --------- |
> |          | GPT-3.5   | ChatGPT   | GPT-4     | GPT-3.5   | ChatGPT   | GPT-4     |
> | Standard | 23.95     | 15.77     | 43.32     | 20.34     | 22.51     | 33.33     |
> | CoT      | 65.04     | 60.72     | 90.36     | 17.31     | 22.51     | 35.06     |
> | Logic-LM | **69.59** | **64.44** | **91.04** | **21.64** | **30.73** | **45.45** |
> | Exe_Rate | 90.9%     | 95.22%    | 99.7%     | 24.24%    | 21.21%    | 31.17%    |
> | Exe_Acc  | 71.33     | 66.16     | 91.01     | 33.93     | 46.93     | 65.28     |
>
> We find that Logic-LM without self-refinement surpasses both Standard and CoT prompting on both datasets across all tested models, including GPT-3.5, ChatGPT, and GPT-4. In the case of the GSM8K dataset, a high *Exe_Rate* is observed, suggesting that the majority of problems within this dataset can be represented as SymPy programs — even though the questions were not originally generated using symbolic reasoners. Conversely, while the *Exe_Rate* for AR-LSAT is somewhat lower, we find that the *Exe_Acc* is significantly greater than that achieved using CoT. This implies that the integration of a symbolic solver, such as Z3 in this case, considerably enhances the model's logical reasoning capabilities when problems can be translated into symbolic language. We argue that these findings, when considered in conjunction with real-world datasets like GSM8K and AR-LSAT, compellingly demonstrate the efficacy of augmenting LLMs with symbolic solvers.
>
> - [1] Zhong et al., AR-LSAT: Investigating Analytical Reasoning of Text. NAACL, 2022.
> - [2] Cobbe et al., Training Verifiers to Solve Math Word Problems. 2021.
>
> ---
>
> > *It is very improper to use these four datasets, and transfer their data back to symbolic language to be solved by symbolic reasoner to illustrate the power of symbolic reasoner.*
>
> In general, we believe studying whether LLMs can transfer these real-world problems into appropriate logical forms is an intriguing direction to explore. There are many related areas for this such as NL-FOL translation, NL-SQL translation, and semantic parsing. In this work, we explore in-context learning as the NL to Logic language translator.
>
> However, we understand the reviewer's concern that transferring a synthetically created dataset back to the symbolic language does not show the effectiveness of our method. Therefore, in the previous replies, we have: 1) clarified that FOLIO is not an automatically created logical reasoning dataset, and 2) demonstrated the effectiveness of LogicLM on two real-world datasets purely created by human experts (GSM8K and AR-LSAT). We hope these could address the reviewer's concern.
>
> ---
>
> > *The performance comparison is established in an unfair experiment --- Logic-LM can use self-refinement to refine the transferred symbolic language, then why "Standard" and "COT" can't use self-refinement as well to directly solve the original problem? What if "Standard" and "COT" use self-refinement to directly solve the original problems?*
>
> We are afraid that this is a misunderstanding. In our Logic-LM model, self-refinement is possible because the symbolic reasoner serves as a feedback mechanism, allowing the model to self-correct based on error messages. This enables the model to refine its generated logic formulations iteratively.
>
> Conversely, "Standard" and "CoT" models lack an analogous feedback mechanism during the inference stage. Once these models produce an answer, there's no clear way to validate its correctness, leaving us without a clear path for correction. Differently, in Logic-LM, the error message from symbolic solver provides a clear feedback signal for whether the symbolic form is correct and what error it contains. This demonstrates another advantage of incorporating a symbolic solver into Logic-LM: it not only enhances reasoning but also facilitates self-refinement through automated feedback.
>
> One way we could implement self-correction for "Standard" and "COT" is to use an external model as the feedback model. Existing methods like Self-Refine [3] and Reflexion [4] use the LLM itself for feedback. After generating an answer, the model is prompted with a question such as, "Is this answer/reasoning correct?" If the model replies "no," it attempts to generate a new answer. However, this is inherently limited by the model's own capabilities. If the LLM cannot solve a particular problem, it is unlikely to accurately assess the correctness of its own answers or reasoning for that problem.
>
> It is also feasible to use an external model to verify the plausibility of each reasoning step, but this introduces a new set of challenges. Verifying the correctness of language-based reasoning is a complex and as-of-yet unresolved research issue. In contrast, the logic language enables more straightforward and reliable validation.
>
> Given the discussions above, incorporating self-refinement is inherently more applicable to Logic-LM than to the "Standard" and "CoT" models. Nonetheless, in response to the reviewer's suggestion, we conducted an additional experiment where the LLM itself serves as the feedback model. Consistent with our expectations, this approach resulted in only marginal performance changes on the FOLIO dataset, as detailed in the table below.
>
> | Language Model | Without Self-refinement | With Self-refinement |
> | -------------- | ----------------------- | -------------------- |
> | GPT-3.5        | 57.84                   | 58.33                |
> | ChatGPT        | 57.35                   | 57.84                |
> | GPT-4          | 70.58                   | 70.09                |
>
> Moreover, even without self-refinement, Logic-LM outperforms "Standard" and "COT" by an obvious margin on GSM8K and AR-LSAT, as shown by our additional results in the previous reply.
>
> - [3] Madaan et al., 2023. Self-refine: Iterative refinement with self-feedback.
>
> - [4] Shinn et al., 2023. Reflexion: Language agents with verbal reinforcement learning.
>
> ---
>
> > *What is the performance of proposed Logic-LM on Entailmentbank dataset? How is it compared to "Standard" and "COT"?*
>
> Entailmentbank is a question-answering dataset that focuses on multi-hop, commonsense reasoning and explanation generation.   However, our paper is focusing on enhancing LLM's ability in solving logical reasoning problems. Therefore, evaluating on Entailmentbank seems out of the scope of this paper. As we admitted in the limitation section, Logic-LM relies on translating reasoning problems into logical formats that can be tackled by symbolic solvers. As a consequence, the model’s applicability is inherently bounded by the expressiveness of the symbolic solver. There are certainly problems that cannot be encoded as logical forms, such as the questions in Entailmentbank. Despite this limitation, the value of integrating symbolic solvers with LLMs should not be discounted. We have already showcased the efficacy of Logic-LM across three real-world datasets: FOLIO, GSM8K, and AR-LSAT. Furthermore, the diversity of existing symbolic solvers—including tools for symbolic mathematics, theorem proving, and logic programming—provides a broad range of utilities that significantly enhance the logical reasoning capabilities of LLMs.
>
> ---
>
> > *What can I learn from this paper given my previous analysis of this paper?*
>
> While we can't determine what the reviewer has learned from our paper, we believe that addressing the reviewer's concerns with our previous replies should clarify the significance of our contributions. To succinctly outline these: 1) We pioneered the integration of symbolic solvers with LLMs using in-context learning for tackling logical reasoning tasks, 2) We developed Logic-LM, a flexible and effective framework for harmonizing various types of symbolic solvers with LLMs, thereby augmenting their logical reasoning capabilities, 3) We explored the self-refinement mechanisms within LLMs for improving logical forms, and 4) We established a foundation with Logic-LM that we anticipate will inspire future neuro-symbolic research, bridging modern LLMs with traditional symbolic logic.

---

### Official Review · Reviewer_TXB8 · 2023-08-21

**Soundness:** 4

**Excitement:**

3: Ambivalent: It has merits (e.g., it reports state-of-the-art results, the idea is nice), but there are key weaknesses (e.g., it describes incremental work), and it can significantly benefit from another round of revision. However, I won't object to accepting it if my co-reviewers champion it.

**Paper Topic And Main Contributions:**

The paper studies logical reasoning task problems. The proposed approach is a tool-augmented LLM.
LLM struggles in making correct reasoning. With the help of specialized symbolic reasoners, LLM can deliver more faithful results on logical reasoning tasks.

**Reasons To Accept:**

- The idea of utilizing symbolic reasoner to help LLM resolver logical reasoning is making sense, to achieve both accuracy in terms of reasoning results and generality of understanding various logical queries.
- The motivation to design three stages or technical components for the overall approach is well explained. LLM is mainly serve as a interpreter to translate text into symbolic formulas, and translate the symbolic results into textual answer.

**Reasons To Reject:**

- The experiments can be further improved by adding necessary compared methods. Are there no few-shot methods/approaches for logical reasoning tasks? What if using some off-the-shelf text-to-symbolic formulator? Or what would be the performance of trained small models on these datasets?
- I appreciate the authors' efforts on curating specific symbolic language grammar for specific type of reasoning task. When conducting experiments, will Logic-LM be prompted with the correct few-shot examples using the most effective grammar? Would that be considered as a data leakage, as in the real-world setting models can never know the exact type of reasoning task and prepare the best few-shot example with best grammars?

**Reproducibility:**

4: Could mostly reproduce the results, but there may be some variation because of sample variance or minor variations in their interpretation of the protocol or method.

**Reviewer Confidence:**

4: Quite sure. I tried to check the important points carefully. It's unlikely, though conceivable, that I missed something that should affect my ratings.

**Typos Grammar Style And Presentation Improvements:**

- I suggest to put Sec 3.2 as a subsection of Sec 3.1. Also Section 3.4 can be aggregate into Sec 3.1. Then Section 3 will have three subsection, each corresponding to one technical component of the propose approach. Now, it is a bit confused to have Symbolic Language Grammar and Self-Refiner, parallel to the three components.

---

> ### Author Rebuttal · Authors · 2023-08-27
>
> We appreciate the valuable insights behind the comments and also want to clarify a few points.
>
> > *The experiments can be further improved by adding necessary compared methods. Are there no few-shot methods/approaches for logical reasoning tasks? What if using some off-the-shelf text-to-symbolic formulator? Or what would be the performance of trained small models on these datasets?*
>
> Thanks for the reviewer's suggestions. In the Related Work section (Lines 144-155), we clarified why traditional neuro-symbolic methods were not included for comparison. These methods: 1) rely on hand-crafted or specialized module designs which are hardly generalizable to other datasets, and 2) they often require large numbers of training data, making them unsuitable for few-shot settings, which is our focus. For this reason, we chose to compare Logic-LM with modern LLMs, such as GPT-4 and GPT-4 + CoT, which are more generalizable and powerful baselines.
>
> Regarding off-the-shelf text-to-symbolic formulators, existing models mainly target SQL or SPASQL formats, which are not applicable to the logical reasoning tasks we study. As far as we're aware, no pre-trained formulators exist for the symbolic languages like FOL, logical programming, or CSP that we employ. This led us to define our own symbolic grammar and use in-context learning for formulation.
>
> We appreciate the reviewer's suggestion on comparing with trained small models. In response to this, we conduct additional experiments on fine-tuning FLAN-T5-XL model to compare with Logic-LM. For a fair comparison, we fine-tune FLAN-T5 on the same $N$ few-shot examples we used in the Logic-LM prompt. The results are shown in the table below.
>
> | Method / Dataset      | ProntoQA | ProofWriter | FOLIO | LogicalDeduction |
> | --------------------- | -------- | ----------- | ----- | ---------------- |
> | FLAN-T5-XL (N=2)      | 49.80    | 34.17       | 34.31 | 28.33            |
> | FLAN-T5-XL (N=50)     | 50.80    | 35.33       | 36.76 | 31.67            |
> | Logic-LM (w. GPT-3.5) | 93.20    | 70.11       | 61.76 | 67.66            |
>
> When trained with the same number of few-shot examples (N=2), it's hard for FLAN-T5 to learn anything useful, with performance close to random guessing across all datasets. Therefore, we relax the constraint by using $N=50$ in-domain examples for training. While this led to some performance improvement, FLAN-T5-XL still lagged significantly behind Logic-LM. These results demonstrate the superior data efficiency and effectiveness of applying in-context learning for problem formulation.  We will include these additional results and findings in the camera-ready version.
>
> ---
>
> > *I appreciate the authors' efforts on curating specific symbolic language grammar for specific type of reasoning task. When conducting experiments, will Logic-LM be prompted with the correct few-shot examples using the most effective grammar? Would that be considered as a data leakage, as in the real-world setting models can never know the exact type of reasoning task and prepare the best few-shot example with best grammars?*
>
> Thanks for asking this question. In our experiments, the grammar for each dataset was predetermined, since we primary focus on accurate problem-to-symbolic-form translation. As an experimental setting, we don't think this is data leakage because it aligns with the premises of few-shot learning where some domain-specific information is assumed to be known. This is analogous to the Chain-of-Thought method, where LLMs are prompted with a few in-domain examples, together with human annotated correct reasoning chains. In Logic-LM, we provide a few correct symbolic formulations, which can be regarded as correct reasoning chains written in symbolic language.
>
> **Additional Experiments**. However, we agree with the reviewer's point that automatically selecting an appropriate symbolic system is an important practical concern. To address this, we conducted an additional experiment to evaluate the potential of using LLMs as grammar designators. We prompted GPT-4 with 8 demonstrations, each consisting of a problem and its corresponding symbolic language, such as `(Problem from FOLIO, "FOL representation")`. These demonstrations were randomly drawn from each of the four datasets we used (two examples per dataset).  Subsequently, we evaluated GPT-4's performance on the grammar designation task to assign new examples to the appropriate symbolic grammar (3-class classification since we have defined three different grammars). Out of 400 randomly sampled test cases (100 from each dataset), GPT-4 achieved a high F1 score of 96.5%. However, we note that this high performance may partially due to the fact that the problems of the four datasets we used are easy to differentiate. In real-world applications, determining the most fitting symbolic system could be more challenging. We will include these new findings and discussions in a discussion section of the camera-ready version. We thank the reviewer for raising this interesting and valuable question.
>
> ---
>
> > *I suggest to put Sec 3.2 as a subsection of Sec 3.1. Also Section 3.4 can be aggregate into Sec 3.1. Then Section 3 will have three subsection, each corresponding to one technical component of the propose approach. Now, it is a bit confused to have Symbolic Language Grammar and Self-Refiner, parallel to the three components.*
>
> We agree with the reviewer's suggestion to reorganize Section 3 into three sub-sections—Problem Formulator, Symbolic Reasoner, and Self-Refiner—which align more coherently with the general framework illustrated in Figure 1. This restructuring will provide a clearer presentation of each technical component of our proposed approach, thereby improving the paper's readability and logical flow. We appreciate this insightful recommendation and will implement it in the final version.

---

### Meta-Review · Area_Chair_bFXo · 2023-09-17

**Recommendation:** 4

**Metareview:**

This paper proposed to improve the reasoning capability of LLMs by giving them the access to a symbolic reasoner. A self-refinement mechanism instructs the LLM to refine the incorrect logical form, by prompting it with the erroneous logic form, the solver’s error message, and a set of demonstrations showing common error cases and their remedies.
Experiment is conducted by transferring four existing synthetic natural language datasets back to symbolic sentences and solving it with a symbolic reasoner. The result shows significant improvement over standard and CoT settings for 3 GPT models.

Strength:
1. The three stage design and self-refinement are reasonable and effective, even though not completely novel.
2. A thorough analysis on the performance of 4 tasks and 3 GPT models.

Weakness:
1. The datasets are synthetic and it is not clear how much impact the proposed approach has on more natural tasks such as Entailmentbank.
2. It is not clear what possible applications can the FOL representation be applied to.

---

### Decision · Program_Chairs · 2023-10-07

**Decision:**

Accept-Findings

**Comment:**

This paper proposed to improve the reasoning capability of LLMs by giving them the access to a symbolic reasoner. A self-refinement mechanism instructs the LLM to refine the incorrect logical form, by prompting it with the erroneous logic form, the solver’s error message, and a set of demonstrations showing common error cases and their remedies.
Experiment is conducted by transferring four existing synthetic natural language datasets back to symbolic sentences and solving it with a symbolic reasoner. The result shows significant improvement over standard and CoT settings for 3 GPT models.

Strength:
1. The three stage design and self-refinement are reasonable and effective, even though not completely novel.
2. A thorough analysis on the performance of 4 tasks and 3 GPT models.

Weakness:
1. The datasets are synthetic and it is not clear how much impact the proposed approach has on more natural tasks such as Entailmentbank.
2. It is not clear what possible applications can the FOL representation be applied to.